# The impact of coal miners' emotions on unsafe behavior: A study on the mediated effects with a moderating role

**Guangjin Chen**[1,2], **Wenliang Xia**[1]*, **Wensheng Wang**[1]

**1** School of Management, China University of Mining and Technology-Beijing, Beijing, China,
**2** Zhengzhou Coal Industry (Group) Co., Ltd., Zhengzhou, China

* wenliang135.xia@gmail.com

**Data availability statement:** All relevant data are within the paper and its Supporting information files.

## Abstract

This study aims to reveal how emotions influence Unsafe Behavior among coal miners, addressing the increasingly severe safety issues in the coal mining industry. As the complexity of mining environments grows, the impact of workers' emotional states on safety behaviors has garnered widespread attention. Based on emotion regulation theory and psychological mediation models, we analyzed survey data from 250 workers across multiple subsidiary mining units of a single coal mining enterprise in China, employing regression analysis and Bootstrap methods to examine the relationships among emotions, unsafe psychological states, and Unsafe Behavior. Additionally, we introduced safety climate as a moderating variable to enhance the explanatory power of the model. The results indicate that positive emotions significantly reduce the occurrence of Unsafe Behavior by lowering unsafe psychological states, whereas negative emotions significantly increase Unsafe Behavior by enhancing unsafe psychological states. Furthermore, unsafe psychological states play a mediating role between emotions and Unsafe Behavior, highlighting the crucial role of psychological factors in the emotional influence on behavior. Further analysis shows that safety climate moderates the relationship between negative emotions and unsafe psychological states. Specifically, under a high safety climate, the impact of negative emotions on unsafe psychological states is weakened. This study provides theoretical support and practical reference for safety management in the coal mining industry, offers empirical evidence for the development of emotional regulation and psychological intervention strategies, and emphasizes the importance of fostering a favorable safety climate.

## Introduction

### Research background

The coal industry is a vital component of China's energy structure, playing a key role in ensuring energy supply and economic development. However, safety issues in coal mines have long constrained the industry's growth, with Unsafe Behavior among coal miners remaining a primary cause of mining accidents and safety incidents [1]. From the perspective of

**Funding:** The author(s) received no specific funding for this work.

**Competing interests:** The authors have declared that no competing interests exist.

safety engineering, the occurrence of accidents is attributed to the combined effects of unsafe conditions and Unsafe Behavior. Heinrich's "88:10:2" rule states that out of 100 accidents, 88 are solely caused by human factors, 10 result from the interaction between human factors and unsafe conditions, and only 2 are due to unforeseen and unpreventable circumstances. Research indicates that coal miners work in high-risk and high-pressure environments, making them susceptible to emotional fluctuations. Negative emotions, particularly (such as stress and anger), can significantly affect individuals' decision-making and behaviors, leading to a decline in safety awareness, which in turn increases unsafe psychological states and triggers Unsafe Behavior [2].

Currently, safety management in coal mines primarily focuses on behavioral norms and technical prevention measures, with insufficient attention paid to miners' emotions and their potential impact on Unsafe Behavior [3]. Emotional issues, especially negative emotions, may indirectly lead to Unsafe Behavior by affecting miners' psychological states (such as complacency and reactance) [4]. For instance, anxiety may cause miners to excessively worry about potential risks, thereby affecting their safety performance [5]. Therefore, studying the emotions of coal miners and their impact on Unsafe Behavior, particularly exploring the mediating role of unsafe psychological states, is significant. This research aims to analyze the relationship between coal miners' emotions and Unsafe Behavior, investigating the mediating role of unsafe psychological states in this process. The goal is to provide theoretical support and practical guidance for coal mine safety management, ultimately reducing accident rates and enhancing safety production levels in coal mining.

## Contribution

This study addresses the gaps in the existing literature regarding the relationship between emotions and Unsafe Behavior among miners, aiming to explore the influencing factors in this field by proposing unsafe psychology as a mediating variable. The specific innovations are as follows:

1. This study systematically investigated the influence of emotions on unsafe behavior and, for the first time, incorporated both positive and negative emotional dimensions in the context of the mining workforce. This dual-dimensional emotional framework constitutes a key innovation of the present research. The findings reveal that different types of emotions lead to significantly distinct behavioral outcomes. By comprehensively examining the mechanisms through which both positive and negative emotions operate, this study further enriches the theoretical understanding of emotional variables in the field of safety behavior research.

2. By incorporating "Unsafe Psychology" as a mediating variable between emotions and unsafe behavior, this study constructs a systematic model and conducts empirical analysis, thereby overcoming the limitations of previous research that primarily focused on external behaviors or isolated emotional factors. Through the identification of this mediation mechanism, the study not only enriches the theoretical pathway by which emotions influence safety behavior but also offers a solid theoretical foundation and practical guidance for the development of psychological intervention–oriented safety management strategies.

3. For the first time, this study introduces safety climate as a moderating variable into the path model between emotions and unsafe psychological states, systematically examining its buffering effect in the transformation from negative emotions to unsafe psychological conditions. This design not only extends the theoretical connotation of safety climate but also further reveals its essential role in high-risk industries, where the construction of an organizational safety culture contributes significantly to individuals' psychological regulation and

behavioral control. The findings provide novel theoretical insights and empirical evidence to inform the development of targeted and preventive safety management interventions.

## Theoretical analysis and research hypotheses

### Emotions and unsafe behavior

Emotion is a complex psychological response that an individual experiences in reaction to internal and external stimuli in specific situations. It typically accompanies physiological changes and behavioral expressions. Emotions involve not only the individual's subjective experience but also physiological responses (such as increased heart rate, sweating, etc.) and behavioral reactions (such as smiling, crying, etc.) [6]. Emotions can be categorized into basic emotions (such as happiness, sadness, anger, fear, disgust, and surprise) and complex emotions (such as shame, guilt, jealousy, etc.) [7]. The emergence of emotions is often closely related to an individual's cognitive appraisal, which refers to the interpretation and judgment of a particular event or situation [8]. Emotions are not only responses to external stimuli but are also influenced by an individual's experiences, cultural background, and social environment [9]. They play a crucial role in regulating behavior, facilitating social interactions, and enhancing adaptability for survival, especially within the unique occupational group of coal miners, where emotional states are particularly crucial. Due to the harsh working conditions and the specific management structure of coal mines, miners' emotions are significantly influenced by external pressures [10]. Research indicates that negative emotions, such as anxiety, fear, and depression, significantly increase Unsafe Behavior in the workplace. For instance, studies have found that under negative emotional states, the incidence of Unsafe Behavior among miners rises significantly, accompanied by a higher risk of behavioral deviations [11]. Moreover, factors such as workload, perceived unfair management, and family pressures can lead to emotional fluctuations among miners, further diminishing their attention and focus, thereby increasing the risk of operational errors and accidents [12]. In contrast, positive emotions exhibit a protective effect. The Emotion Maintenance Hypothesis suggests that in high-risk environments, positive emotions can help workers better avoid risks and reduce the occurrence of Unsafe Behavior [13].

Based on this, the following hypotheses are proposed:

Hypothesis H1: Positive emotions of miners have a significant negative effect on their Unsafe Behavior.

Hypothesis H2: Negative emotions of miners have a significant positive effect on their Unsafe Behavior.

### Emotions and unsafe psychological states

Emotions, as a psychological state, have a profound impact on an individual's behavior and cognitive processes, especially in high-risk environments. In such contexts, emotions can be a significant factor contributing to unsafe psychological states among miners. The emotional state of miners not only influences their perception of the work environment but also affects their safety awareness and psychological responses, potentially leading to Unsafe Behavior [14]. Understanding these dynamic relationships is crucial for improving safety management measures. This section will explore the impact of emotions—both positive and negative—on unsafe psychological states and will propose corresponding hypotheses.

Negative emotions, particularly anxiety, stress, and fear, are widely recognized as key factors triggering unsafe psychological states. Research indicates that when individuals are in a negative emotional state, they are more likely to develop pessimistic cognitions about their

work environment, thereby increasing the likelihood of unsafe psychological states [15]. For instance, anxiety may lead miners to excessively worry about potential risks, which can subsequently affect their adherence to safety protocols.

Hypothesis H3: Negative emotions among miners have a significant positive effect on their unsafe psychological states.

Positive emotions, such as happiness, satisfaction, and relaxation, typically enhance individuals' mental health levels and strengthen their sense of safety and self-protection awareness [16]. Miners in a positive emotional state typically exhibit a stronger tendency to avoid risks and are better able to assess potential dangers in their work environment, thereby reducing the occurrence of unsafe psychological states. Research shows that positive emotions can also promote teamwork and communication, further enhancing safety awareness. As suggested by the emotion maintenance hypothesis, positive emotions can improve miners' perceptions of their work environment, thereby increasing their sense of safety. This emotional state not only helps to enhance individual work efficiency but can also significantly reduce the risk of accidents.

Hypothesis H4: Positive emotions among miners have a significant negative effect on their unsafe psychological states.

## Unsafe psychological states and unsafe behavior

Unsafe psychological states refer to the negative mental states individuals experience when facing potential dangers in their work environment, including neglect of safety regulations, underestimation of risks, or resistance to safe behaviors. Research indicates a close relationship between unsafe psychological states and Unsafe Behavior; unsafe psychological states not only affect individuals' safety awareness but can also alter workers' decision-making and behavioral patterns, leading to the occurrence of Unsafe Behavior [17]. Therefore, understanding how unsafe psychological states influence Unsafe Behavior is crucial for reducing Unsafe Behavior among miners and enhancing workplace safety.

Unsafe psychological states often manifest as miners' neglect of potential risks, disregard for safety procedures, and underestimation of dangers. This psychological state frequently leads miners to overlook safety requirements, increasing the likelihood of Unsafe Behavior. Studies have found that individuals' unsafe psychological states directly impact their compliance with safety regulations, subsequently resulting in Unsafe Behavior [18]. Thus, unsafe psychological states are an important factor influencing miners' behaviors.

Hypothesis H5: There is a significant positive relationship between miners' unsafe psychological states and Unsafe Behavior.

## Mediating effects of unsafe psychological states

According to individuals' cognitive decision-making processes, emotional states play a crucial role in psychological and behavioral decisions. Individuals form their perceptions of the environment through the recognition and processing of emotions, and these perceptions guide their behavioral decisions to some extent, providing feedback. In the work environment of miners, emotional states not only influence their perception of safety risks but also affect their behavioral performance through this perception. In other words, emotions may indirectly influence miners' safety behaviors by affecting their psychological states [19].

Based on the research, this study posits that miners' emotional states have an indirect effect on Unsafe Behavior through unsafe psychological states. Negative emotions (such as anxiety and stress) may trigger miners' unsafe psychological states (such as complacency and

reactance), affecting their perception of potential risks and leading to Unsafe Behavior. Conversely, positive emotions may effectively reduce Unsafe Behavior by decreasing the occurrence of unsafe psychological states.

Hypothesis H6: Unsafe psychological states mediate the relationship between positive emotions and Unsafe Behavior.

Hypothesis H7: Unsafe psychological states mediate the relationship between negative emotions and Unsafe Behavior.

### Moderating role of safety climate

A safety climate refers to the degree of emphasis on safety within the work environment and the safety culture established within the organization. This climate acts on an organizational psychological level by Safety climate refers to the degree of emphasis on safety within the work environment and the safety culture that is established within an organization. This climate acts on an organizational psychological level by influencing employees' perceptions of potential risks in the work environment and their attitudes towards safety behavior. Research indicates that safety climate can affect the impact of emotions and psychological states. In a high safety climate, employees are more likely to recognize the importance of safety, thereby reducing the influence of negative emotions on unsafe psychology [20].

In the coal mining work environment, a high level of safety climate not only enhances miners' attention to safety procedures but also increases their psychological resilience when facing negative emotions, thereby reducing the likelihood of unsafe behavior. As an external organizational cultural factor, safety climate intervenes in the pathway by which negative emotions affect unsafe psychology. Therefore, understanding the moderating role of safety climate is crucial for developing effective safety management strategies.

Hypothesis H8: Safety climate has a significant moderating effect between negative emotions and unsafe psychology.

### Conceptual model construction

Based on the theoretical foundation and research hypotheses, a systematic structural model has been constructed, as shown in Fig 1. This theoretical model illustrates the hypothesized relationships among emotions, safety climate, unsafe psychological states, and unsafe behavior. Specifically, positive and negative emotions are treated as independent variables that influence individuals' unsafe psychological states and unsafe behavior. Unsafe psychological states serve as a mediating variable, acting as a bridge between emotions and behavior.

Additionally, the moderating role of safety climate is incorporated into the model, specifically moderating the pathway from negative emotions to unsafe psychological states. By analyzing this structural model, we can gain a deeper understanding of how emotions impact miners' safety behaviors and reveal how safety climate can modulate the relationship between emotions and psychological states.

## Materials and methods

### Participants and procedure

This study was conducted in July 2024 using a cluster random sampling method. A total of 250 male coal miners from mining units affiliated with a single coal mining enterprise, located in Shandong, Henan, Shanxi, and Inner Mongolia, were selected as research participants. The purpose of this study was to investigate the psychological states and safety behaviors of front-line workers in high-risk mining environments.

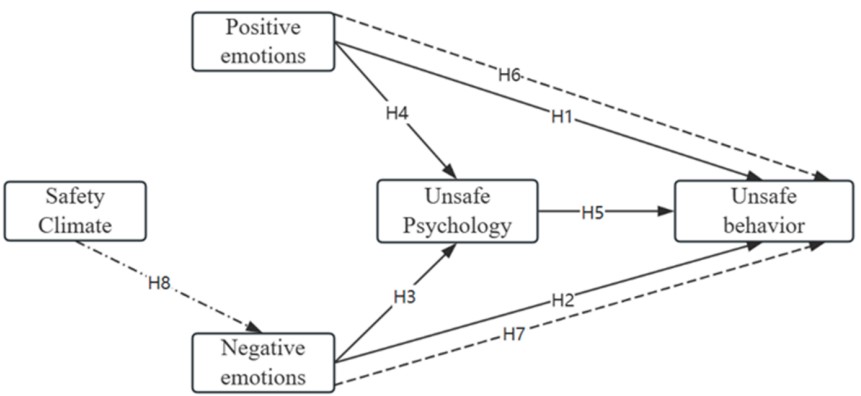

**Fig 1. Conceptual model.**

Front-line coal miners are the core labor force in coal production. They work deep underground and are exposed to various potential safety risks, including gas explosions, collapses, and equipment failures. These high-risk factors not only threaten workers' physical safety but also have profound impacts on their mental health. Prolonged exposure to tense and uncertain working environments can lead to emotional issues such as anxiety and depression, thereby affecting their work efficiency and safety behavior.

The research team designed a comprehensive questionnaire tailored to the unique characteristics of the coal mining industry. To ensure the authenticity and reliability of the data, the survey was conducted anonymously. Participants completed the questionnaire during shift changes underground deliberate choice considering the intensity and specificity of coal mining operations. During shift transitions, miners' emotions and mental states are influenced by the work environment, peer interactions, and upcoming tasks, making it a valuable period to capture both psychological and physiological pressures.

Demographic results show that most participants were married men aged between 40 and 50, with the majority having completed high school or vocational education. Most were temporary workers and reported being in good health, reflecting the typical characteristics of front-line coal miners in China. The sample thus provides a solid demographic foundation for subsequent analysis.

A summary of participants' demographic characteristics is presented in Table 1.

### Measures

1. Positive Affect and Negative Affect Scale (PANAS): Developed by Watson et al., this scale consists of two dimensions: positive emotions and negative emotions [21]. This study integrates findings from domestic and international scholars on safety culture, considering the specific national context and psychological conditions of coal miners, with a particular focus on their emotional states. The scale is based on the Chinese version revised by Qiu Lin et al., assessing participants' positive and negative emotions over the past week [22]. The scale contains a total of 20 items (e.g., "happy," "proud"), evaluated using a five-point Likert scale with options ranging from "strongly disagree" to "strongly agree." The internal consistency (Cronbach's alpha) of the questionnaire in this study was found to be 0.93.

2. Unsafe Psychological Questionnaire for Miners: This questionnaire, developed by Naiwen Li, consists of 20 items and uses a five-point scoring system, where higher scores

**Table 1. Description of sample characteristics distribution.**

| Variable | Option | Frequency | Percentage |
|---|---|---|---|
| Age Structure | 18–30 years | 9 | 3.6% |
| | 30–40 years | 58 | 23.2% |
| | 40–50 years | 116 | 46.4% |
| | 50–60 years | 67 | 26.8% |
| Marital Status | Single | 4 | 1.6% |
| | In a relationship (unmarried) | 1 | 0.4% |
| | Married | 235 | 94.0% |
| | Divorced | 5 | 2.0% |
| | Remarried | 5 | 2.0% |
| Educational Background | Primary School | 8 | 3.2% |
| | Junior High | 49 | 19.6% |
| | High School/Vocational | 148 | 59.2% |
| | Associate Degree | 34 | 13.6% |
| | Bachelor's Degree or higher | 11 | 4.4% |
| Employment Type | Formal Worker | 31 | 12.4% |
| | Temporary Worker | 219 | 87.6% |
| Health Status | Good Health | 230 | 92.0% |
| | Average Health | 20 | 8.0% |

indicate more severe unsafe psychological states [23]. The questionnaire includes four factors: complacency, reactance, feelings of helplessness regarding safety, and temporary psychological states. The homogeneity reliability for these four factors is 0.754, 0.794, 0.876, and 0.823, respectively, with an overall reliability of 0.859, indicating high credibility.

3. Unsafe Behavior Questionnaire: The design of the miners' unsafe behavior scale is based on the relatively mature unsafe behavior scale by Neal and Griffin [24]. Feedback from numerous studies that have used this scale multiple times indicates that it has good reliability and validity. This study measures Unsafe Behavior among coal miners based on the findings of Neal and Griffin.

4. Safety Climate Questionnaire

Different researchers have developed various safety climate scales for different industries. Zohar initially created an eight-dimension questionnaire for the manufacturing industry, while others, such as Isla and Diaz, and Lee, have developed their own scales in different fields [25]. However, due to industry differences, these foreign scales cannot be directly applied to Chinese coal mining enterprises. Therefore, this study adopts and adjusts the safety climate scale developed specifically for the coal mining industry by Ye Xinfeng to better reflect the actual situation [26].

## Data analysis

Data analyses were conducted using SPSS 27.0. Internal consistency reliability for the main variables, including positive emotion, negative emotion, unsafe psychological state, unsafe behavior, and safety climate—was assessed using Cronbach's alpha coefficients. Descriptive statistics and Pearson correlation analyses were performed to examine the distributional characteristics and bivariate relationships among the variables.

To test the core hypotheses, multiple regression analyses were conducted to examine the direct effects of emotional variables on unsafe psychological states and unsafe behavior. Mediation effects were assessed using the bias-corrected percentile bootstrap method with 5,000 resamples to generate 95% confidence intervals, evaluating whether unsafe psychological states mediated the relationship between emotions and unsafe behavior.

Additionally, moderated mediation was tested using PROCESS macro (Model 7), focusing on the moderating role of safety climate in the relationship between negative emotion and unsafe psychological states. All parameter estimates were calculated using the maximum likelihood method (ML), and missing data were handled via SPSS default procedures to ensure robustness.

### Ethics statement

This study was conducted in accordance with the Declaration of Helsinki and was approved by the Ethics Committee of the School of Management, China University of Mining and Technology-Beijing on May 1, 2024 (Approval No. CUMTB-202405001). Data collection began on July 1, 2024, and was completed on July 10, 2024.

1. Informed Consent: All participants were clearly informed about the study's basic information, purpose, procedures, and other important considerations before joining the study. Participants reviewed and agreed to the informed consent form before completing the questionnaire and were informed that they could withdraw from the study at any time during participation.

2. Anonymity and Confidentiality: Participants' privacy and data confidentiality were strictly protected during data collection. All collected data were anonymized during analysis, with no personal identifying information recorded. Only the research team has access to the data, which is used solely for academic analysis within this study.

3. Psychological Support: If participants experience any discomfort or psychological distress during or after completing the questionnaire, the research team will provide necessary support and guidance or assist in contacting relevant professional institutions.

This study has been approved by the relevant ethics committee and is committed to the highest ethical standards. The informed consent form includes the research team's contact phone number and email address. If you have any ethical inquiries or suggestions, you are welcome to contact the research team at any time.

In summary, this study, through rigorous statistical methods and analytical techniques, aims to provide an in-depth understanding and scientific basis for the relationship between trait anxiety and unsafe behavior among coal miners, offering valuable insights for subsequent intervention measures and policy formulation.

## Results

### Reliability analysis

In this study, reliability analysis was conducted using SPSS 27.0 statistical analysis software to test the reliability of the four variables: emotion, safety climate, unsafe psychology, and unsafe behavior, along with their dimensions. The results are shown in Table 2.

**Table 2**. **Results of reliability analysis for each scale.**

| Variable | Alpha | number |
|---|---|---|
| Positive Emotion | 0.944 | 10 |
| Negative Emotion | 0.939 | 10 |
| Unsafe Psychology | 0.969 | 20 |
| Unsafe Behavior | 0.970 | 10 |
| Safety Climate | 0.920 | 7 |

From the table, it can be observed that the Cronbach's $\alpha$ values for emotion, safety climate, unsafe psychology, and unsafe behavior are all greater than 0.7, indicating that the questionnaires have good internal consistency and high reliability. This also suggests that using these subscales for research on the relationships between the variables is reliable.

## Correlation analysis

As shown in Table 3, there were significant correlations among emotions, safety climate, unsafe psychological states, and unsafe behaviors. Positive emotions were negatively correlated with both unsafe psychological states and unsafe behaviors, whereas negative emotions showed significant positive correlations with these two variables. Meanwhile, safety climate exhibited positive associations with multiple variables. In particular, the correlation coefficient between unsafe psychological states and unsafe behaviors reached 0.857, indicating a strong relationship and providing a solid basis for the subsequent mediation analysis.

## Main effects testing

Table 4. presents the results of the multiple regression analysis on the main effects of emotional variables on unsafe behavior and unsafe psychology. This section will provide an academic description from three aspects: control variables, emotional variables, and model fit.

(1) Effects of Control Variables

Regarding control variables, age was found to be a significant negative predictor of unsafe behavior in Models M1 and M2 ($\beta = -0.128^*$, $\beta = -0.131^*$, p < 0.05), indicating that the

**Table 3. Correlation analysis of key measurement dimensions.**

| Variable | M | SD | 1 | 2 | 3 | 4 |
|---|---|---|---|---|---|---|
| Positive Emotions | 44.000 | 6.463 | 1 | – | – | – |
| Negative Emotions | 24.440 | 10.829 | −0.241** | 1 | – | – |
| Safety Climate | 31.760 | 3.877 | 0.626** | −0.242** | 1 | – |
| Unsafe Psychology | 41.460 | 19.055 | −0.256* | 0.625** | −0.327** | 1 |
| Unsafe Behavior | 19.180 | 10.428 | −0.171* | 0.547** | −0.300* | 0.857** |

*Note: p < .05 (\*), p < .01 (\*\*).*

**Table 4. Results of the regression analysis on the impact of emotions on unsafe behavior.**

| Variables | Unsafe Behavior | | | | Unsafe Psychology | | |
|---|---|---|---|---|---|---|---|
| | M1 | M2 | M3 | M4 | M5 | M6 | M7 |
| Age | −0.128* | −0.131* | −0.092 | −0.056 | −0.085 | −0.089 | −0.043 |
| Marital Status | −0.004 | −0.002 | −0.028 | 0.043 | −0.055 | −0.051 | −0.083 |
| Education Level | 0.025 | 0.016 | 0.010 | −0.008 | 0.038 | 0.024 | 0.021 |
| Employment Type | 0.048 | 0.046 | 0.033 | 0.026 | 0.025 | 0.021 | 0.008 |
| Health Status | 0.166** | 0.120 | 0.070 | −0.011 | 0.207*** | 0.134* | 0.096 |
| Positive Emotions | – | −0.134* | – | – | – | −0.212*** | – |
| Negative Emotions | – | – | 0.530*** | – | – | – | 0.609*** |
| Unsafe Psychology | – | – | – | – | 0.857*** | – | – |
| Unsafe Behavior | – | – | – | – | – | – | – |
| $R^2$ | 0.044 | 0.060 | 0.314 | 0.741 | 0.053 | 0.060 | 0.409 |
| $\Delta R^2$ | 0.025 | 0.037 | 0.297 | 0.734 | 0.034 | 0.037 | 0.394 |
| F | 2.268* | 4.071* | 95.697*** | 115.584*** | 2.726* | 4.071** | 146.461*** |

**Note:** p < .05 (\*), p < .01 (\*\*), p < .001 (\*\*\*).

likelihood of unsafe behavior decreases with age. However, the effect of age was not statistically significant in the other models (M3 to M7, p > 0.05). Other control variables, including marital status, education level, and employment type, showed no significant effects on unsafe behavior or unsafe psychological states across the models. Notably, in Models M1 and M5, health condition positively predicted unsafe behavior ($\beta = 0.166^{**}$ and $\beta = 0.207^{***}$, p < 0.01 or p < 0.001), suggesting that poorer health status is associated with a higher risk of unsafe behavior.

(2) Effects of Emotional Variables

The regression results across models demonstrated that emotional variables had significant effects on both unsafe psychological states and unsafe behavior. In Models M2 and M6, positive emotions were negatively associated with unsafe behavior ($\beta = -0.134^*$, p < 0.05) and unsafe psychological states ($\beta = -0.212^{***}$, p < 0.001), supporting Hypotheses H1 and H4. In Models M3 and M7, negative emotions positively predicted unsafe behavior ($\beta = 0.53^{***}$) and unsafe psychological states ($\beta = 0.609^{***}$), supporting Hypotheses H2 and H3. Model M4 further showed that unsafe psychological states had a significant positive predictive effect on unsafe behavior ($\beta = 0.857^{***}$, p < 0.001), supporting Hypothesis H5.

(3) Model Explanatory Power

In terms of model explanatory power, the results of $R^2$ and $\Delta R^2$ indicated that the inclusion of emotional variables significantly improved the models' ability to explain unsafe behavior and unsafe psychological states. For example, in Model M4, $R^2 = 0.741$ and $\Delta R^2 = 0.734$, indicating that unsafe psychological states had strong explanatory power for unsafe behavior. In Model M7, $R^2 = 0.409$ and $\Delta R^2 = 0.394$, demonstrating that negative emotions strongly predicted unsafe psychological states. Furthermore, all models yielded statistically significant F-values (p < 0.05 or p < 0.001), confirming the overall statistical robustness of the regression models.

## Mediation effects test

Using the bias-corrected percentile Bootstrap method with 5,000 samples, the mediation effect of unsafe psychology between negative emotions and unsafe behavior was analyzed. The results are shown in Table 5:

The indirect effect of positive emotions on unsafe behavior through unsafe psychological states was −0.359, with a bootstrap standard error of 0.079 and a 95% confidence interval of [−0.515, −0.207], which did not include zero, indicating a statistically significant mediation effect. Thus, Hypothesis H6 was supported. Similarly, the indirect effect of negative emotions on unsafe behavior via unsafe psychological states was 0.509, with a bootstrap standard error of 0.064 and a 95% confidence interval of [0.383, 0.638]. Since the confidence interval did not contain zero, this mediation effect was also statistically significant, supporting Hypothesis H7.

**Table 5. Results of the mediation effect analysis.**

| Effect Type | Effect Value | Bootse | Bootsetrap95%CI | |
|---|---|---|---|---|
| | | | Lower | Upper |
| Positive Emotions | −0.359 | 0.079 | −0.515 | −0.207 |
| Negative Emotions | 0.509 | 0.064 | 0.383 | 0.638 |

## Moderation effect testing

Following the confirmation of significant mediation effects, Model 7 of the PROCESS macro was employed to examine the moderating role of safety climate in the relationship between negative emotions and unsafe psychological states. As shown in Table 6, the interaction term between negative emotions and safety climate significantly predicted unsafe psychological states ($\beta$ = 0.061, 95% bootstrap confidence interval [0.006, 0.116]), indicating that the interaction effect was statistically significant.

Fig 2 illustrates the pattern of the moderation effect. Across different levels of negative emotions, the group with a high safety climate consistently reported lower levels of unsafe psychological states compared to the low safety climate group. This difference was particularly pronounced under low levels of negative emotions. These results suggest that safety climate played a buffering role in the first stage of the mediation path by moderating the effect of negative emotions on unsafe psychological states.

Taken together, the findings provide support for Hypothesis H8, indicating that safety climate significantly moderates the relationship between negative emotions and unsafe psychological states.

**Table 6. Test of the moderated mediation model.**

| Variables | Unsafe Psychology | | | Unsafe Behavior | | |
|---|---|---|---|---|---|---|
| | $\beta$ | $t$ | 95% CI | $\beta$ | $t$ | 95% CI |
| Negative Emotions | 0.942 | 10.002*** | [0.766, 1.128] | 0.18 | −0.494 | [−0.062, 0.975] |
| Unsafe Psychology | – | – | – | 0.463 | 20.165*** | [0.418, 0.508] |
| Unsafe Climate | -1.041 | -4.177*** | [−4.056, −0.988] | – | – | – |
| Int_1 | 0.061 | 2.165* | [0.006, 0.116] | – | – | – |
| R | 0.659 | | | 0.858 | | |
| R² | 0.435 | | | 0.735 | | |
| F | 63.014*** | | | 343.108*** | | |

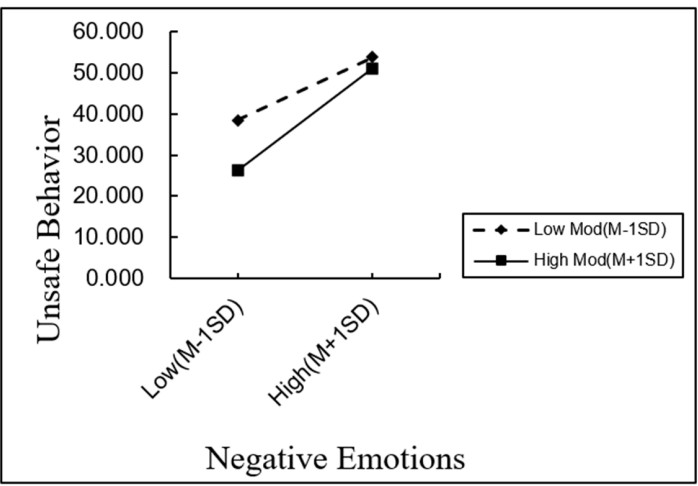

**Fig 2. Moderating effect diagram.**

## Discussion

### Summary of key findings

This study, based on a sample of frontline workers in Chinese coal mining enterprises, constructed and validated a mediation model linking emotions, unsafe psychological states, and unsafe behaviors, while further examining the moderating role of safety climate. The findings provided overall support for the proposed hypotheses, with several key results highlighted below:

First, the results indicated that positive emotions were significantly negatively associated with both unsafe psychological states and unsafe behaviors, whereas negative emotions were significantly positively associated with these outcomes. This suggests a directional difference in how emotional valence predicts safety-related psychological and behavioral responses. Notably, the study confirmed that positive emotions are not merely the opposite of negative emotions but exert independent effects, supporting the dual-pathway emotional framework. While most prior research has focused on the amplifying effects of negative emotions on safety risks, the present findings suggest that positive emotions may serve as a "buffer" or "protective" factor in high-risk work environments, underscoring the theoretical and practical relevance of a bidimensional emotional perspective.

Second, unsafe psychological states were found to mediate the relationship between emotions and unsafe behaviors. Specifically, positive emotions reduced unsafe behaviors by lowering unsafe psychological states, whereas negative emotions increased unsafe behaviors indirectly by amplifying these states.

Finally, the moderated mediation analysis revealed that safety climate significantly moderated the pathway from negative emotions to unsafe psychological states. A higher level of perceived safety climate was found to mitigate the impact of negative emotions on unsafe psychological tendencies, indicating that a supportive organizational environment can effectively buffer individual-level emotional risks.

### Theoretical implications

This study, grounded in an emotional perspective, constructed and validated a theoretical pathway of "Emotions → Unsafe Psychological States → Unsafe Behaviors," while incorporating the moderating role of safety climate. It extends existing research on the psychological mechanisms of safety behavior within high-risk occupational settings.

Previous studies on safety behavior have predominantly focused on contextual factors such as institutional regulations, work environments, and external stressors, while giving limited attention to individuals' internal emotional states. By simultaneously incorporating both positive and negative dimensions of emotion, this study demonstrates that different types of emotions exert opposite effects on unsafe psychological states and behaviors. Importantly, the findings highlight the independent and constructive role of positive emotions, thereby challenging the prevailing risk-focused emotional perspective in prior research.

At the level of mechanism, the study empirically confirmed the mediating role of unsafe psychological states in the relationship between emotion and behavior. This suggests that individuals' subjective perceptions of risk serve as a critical cognitive-affective bridge between emotional responses and behavioral outcomes. This mechanism enriches the psychological explanatory models in safety behavior research and offers theoretical insights into decision-making processes in complex work settings.

In addition, the finding that safety climate significantly buffers the adverse impact of negative emotions on unsafe psychological states emphasizes the critical role of organizational

culture in shaping employees' emotional responses and psychological well-being. By integrating emotional and organizational-level variables, this study advances the application of safety climate theory at the individual level.

Overall, this research, situated within the high-risk context of coal mining, establishes a comprehensive multi-level model that systematically reveals the dynamic interplay among emotion, psychology, and behavior. The findings provide a theoretical extension to safety management research and offer a conceptual foundation for developing cross-level safety intervention strategies.

## Practical implications

The findings of this study suggest that emotions are not only a key factor influencing miners' psychological states but also indirectly affect the occurrence of unsafe behaviors through mediating mechanisms. This provides a novel intervention perspective for safety management in high-risk industries such as coal mining. Current safety management strategies primarily focus on operational regulations and accident prevention, often neglecting the role of employees' emotional fluctuations. The present study highlights the importance for managers to pay close attention to frontline workers' daily emotional states, particularly in identifying and addressing negative emotions in a timely manner to prevent the escalation into unsafe psychological states and behaviors.

Furthermore, the results reveal that positive emotions play a significant protective role by enhancing employees' psychological stability and risk-avoidance capability. Therefore, emotional management should not be limited to the control of negative emotions but should also focus on the stimulation and maintenance of positive emotional experiences. Organizations may achieve this by strengthening positive reinforcement, fostering a supportive organizational climate, and improving the quality of interpersonal relationships. These strategies can enhance employees' overall emotional resources and increase their intrinsic motivation to engage in safe behaviors.

In addition, the study found that a strong safety climate can buffer the negative impact of emotional distress at the organizational level. This implies that an institutionalized culture of safety not only promotes employees' adherence to safety rules but also provides psychological support and behavioral regulation during periods of emotional turbulence. By establishing open communication channels, enhancing employee engagement and sense of belonging, and continuously promoting shared safety values, organizations can further improve their safety climate and reduce the likelihood of unsafe psychological states and behaviors.

Taken together, this study proposes a multi-level intervention framework, ranging from "emotion recognition—psychological regulation—behavioral control" to "organizational support—cultural construction." It underscores the critical role of emotional variables within the behavior management system and offers a practical pathway for developing human-centered safety management strategies.

## Limitations and future research

This study, grounded in an emotional perspective, constructed a mediation model involving unsafe psychological states and verified the moderating role of safety climate, offering a novel theoretical lens for understanding the psychological mechanisms underlying employee behavior in high-risk industries. However, several limitations should be acknowledged and addressed in future research.

First, although the sample covered multiple coal mining sites across Shandong, Shanxi, Henan, and Inner Mongolia, all participants were employed within the same corporate group,

which shares a highly consistent organizational structure and management culture. This homogeneity may limit the generalizability of the findings to other industries or institutional contexts. Future research could expand the sampling scope to include frontline workers from different industries and organizational systems to enhance the applicability and universality of the proposed model.

Second, the study employed a cross-sectional design and relied entirely on self-reported data from employees. Despite efforts to minimize bias during survey administration, issues such as social desirability effects and common method bias cannot be fully excluded, and causal relationships between variables remain uncertain. Future research could incorporate multi-source data, such as supervisor ratings or safety incident records, and adopt longitudinal or experimental designs to strengthen causal inferences and capture dynamic psychological processes more accurately.

In addition, the current model primarily focused on state-level emotions and psychological mechanisms, without accounting for stable personality traits—such as neuroticism, impulsivity, or self-regulatory capacity—that may influence individuals' emotional response patterns. Incorporating individual difference variables into future models could help develop more predictive and theoretically robust frameworks. Furthermore, with advancements in artificial intelligence and emotion recognition technologies, future studies could explore the integration of multimodal emotion detection systems—based on voice, facial expressions, or physiological signals—to objectively and dynamically monitor employees' emotional states. Such approaches may improve the timeliness and accuracy of emotion data and promote a shift from questionnaire-based methods to intelligent sensing in safety behavior research.

Finally, emotional expression and its behavioral consequences may be moderated by cultural norms and institutional environments. Future research is encouraged to validate the stability of the proposed model in cross-cultural contexts and advance comparative studies that examine emotion–psychology–behavior pathways across different cultural settings and local organizational practices.

## Conclusion

This study, based on a sample of coal mine workers, constructed a mechanism linking emotions, unsafe psychological states, and unsafe behaviors, and further incorporated safety climate as a moderating variable. The results systematically revealed the indirect pathway through which emotional factors influence unsafe behavior. The findings indicated that positive emotions exert a significant protective effect by reducing the risk of unsafe psychological states and behaviors, whereas negative emotions increase the likelihood of rule violations by triggering unsafe psychological responses. Moreover, safety climate demonstrated a critical buffering effect within the pathway from negative emotions to unsafe psychological states. These findings not only enrich the theoretical framework concerning the emotion–behavior relationship but also offer more targeted intervention approaches and optimization strategies for safety management in high-risk industries.

## Supporting information

**S1 Data. Raw processed data on emotions and unsafe behavior.**
(XLSX)

## Author contributions

**Project administration:** Guangjin Chen.

**Supervision:** Wensheng Wang.

**Writing – original draft:** Wenliang Xia.

**Writing – review & editing:** Wenliang Xia.

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
