## [Decision Letter · Decision Letter 0]

21 Jun 2025

PONE-D-25-18288The Impact of Coal Miners' Emotions on Unsafe Behaviors: A Study on the Mediating Role of Unsafe Psychological StatesPLOS ONE

Dear Dr. Xia,

Thank you for submitting your manuscript to PLOS ONE. After careful consideration, we feel that it has merit but does not fully meet PLOS ONE’s publication criteria as it currently stands. Therefore, we invite you to submit a revised version of the manuscript that addresses the points raised during the review process.

We look forward to receiving your revised manuscript.

Kind regards,

Damiano GIRARDI

Academic Editor

PLOS ONE

Journal Requirements:

5. Please upload a copy of Figure 2, to which you refer in your text on page 11. If the figure is no longer to be included as part of the submission please remove all reference to it within the text.

7. Please remove all personal information, ensure that the data shared are in accordance with participant consent, and re-upload a fully anonymized data set.

Reviewers' comments:

Reviewer's Responses to Questions

**Comments to the Author**

1. Is the manuscript technically sound, and do the data support the conclusions?

Reviewer #1: Yes

Reviewer #2: Partly

2. Has the statistical analysis been performed appropriately and rigorously? 

Reviewer #1: Yes

Reviewer #2: Yes

3. Have the authors made all data underlying the findings in their manuscript fully available?

Reviewer #1: Yes

Reviewer #2: Yes

4. Is the manuscript presented in an intelligible fashion and written in standard English?

Reviewer #1: Yes

Reviewer #2: Yes

5. Review Comments to the Author

Reviewer #1: The paper aims to explore how emotions influence the unsafe behaviors of coal miners, in response to the increasing safety issues in the coal industry. Based on survey data from 250 workers in multiple coal mining companies in China, it examines the relationships among emotions, unsafe psychological states, and unsafe behaviors using mathematical methods. The topic is interesting. However, there are still several issues that need to be considered before the paper is accepted by Plos One.

1. The authors mention that they have considered both positive and negative emotional dimensions,and that different types of positive and negative situations can lead to significantly different behavioral outcomes. So, how were the dimensions of positive and negative situations divided, and what specific criteria were used?

2. How was the relationship between the three (emotions,unsafe psychological states,and unsafe behaviors) established, and how scientific is it?

3. Emotional fluctuations can increase miners' emotional volatility, further reducing their attention and concentration, thereby increasing the risk of operational errors and accidents. There are many factors that can influence emotions, some of which may be negative while others may be positive. But how should these emotions be understood?

4. Did the authors distinguish between emotions based on coal mine safety management and those based on other factors(including family,society,etc.) in the questionnaire? Although different emotions can have certain relationships with emotions,unsafe psychological states, and unsafe behaviors, emotions related to coal mine safety are only a part of the miners' overall emotions.

5. The paper mentions that a cluster random sampling method was used to select 250 male workers from coal mining companies in Shandong, Henan, Shanxi, and Inner Mongolia as the subjects of the study. However, the conclusion states that the data came from a single coal mining company,which is confusing. I speculate that this might refer to a single coal mining company that includes subordinate coal mines distributed across different provinces.

6. Section 1.2, the first letter of the title is not capitalized.

Reviewer #2: Thank you for giving me this opportunity to review The theoretical model of this paper is innovative. It is the first time to integrate the dual dimensions of positive and negative emotions, the mediating mechanism of unsafe psychology and the moderating role of safety climate in the study of Miners' safety behavior, a multi-level theoretical framework is constructed.However, there are also the following problems: 1) the sample representation and method limitations, the sample size of the data is a bit small, the universality of the conclusion is questionable, and the data presentation and analysis results need to be further modified, the direction of effect of health status on unsafe behaviors was inconsistent in table 3(M1: β = 0.166; M4: β = -0.011) , please explain further. The text mentions Fig. 2, but the manuscript does not provide an illustration, which affects the interpretation of the results.

6. PLOS authors have the option to publish the peer review history of their article (what does this mean?). If published, this will include your full peer review and any attached files.

Reviewer #1: No

Reviewer #2: No

---

## [Author Response · Author response to Decision Letter 1]

9 Jul 2025

Response to Editorial Comments

Dear Editor,

Thank you for your detailed guidance regarding the manuscript formatting. We have carefully revised the manuscript in strict accordance with the official formatting templates provided by PLOS ONE. This includes adjustments to the overall structure, section headings, reference formatting, and file naming conventions, in order to fully comply with all requirements. If there are still any issues, we sincerely welcome further feedback and will address them promptly.

Dear Editor,

Thank you for your careful review. We confirm that the ethics statement has been placed exclusively in the “Materials and Methods” section (Chapter 3) of the manuscript, as required, and does not appear in any other part of the text.

3. Please update your submission to use the PLOS LaTeX template. /latex。

Dear Editor,

Thank you for your kind reminder. We have reformatted and revised the manuscript using the official PLOS LaTeX template, ensuring that all sections comply with the journal’s submission guidelines. If there are any remaining issues, please kindly let us know, and we will make further improvements accordingly.

Dear Editor,

Thank you for your kind reminder. We have confirmed that the title in the online submission system is now consistent with the title in the manuscript. If there are still any discrepancies, please let us know, and we will correct them immediately.

5. Please upload a copy of Figure 2, to which you refer in your text on page 11. If the figure is no longer to be included as part of the submission please remove all reference to it within the text.

Dear Editor,

Thank you for your review comments. We have re-uploaded Figure 2 and ensured that its citation and numbering are consistent within the manuscript.

6. Please include captions for your Supporting Information files at the end of your manuscript, and update any in-text citations to match accordingly.

Dear Editor,

Thank you for your kind reminder. We confirm that no additional Supporting Information files were uploaded with this submission, as all essential content has been fully presented within the main text. Therefore, no separate Supporting Information section has been added at the end of the manuscript. If there are specific formatting requirements, please kindly let us know and we will make the necessary adjustments promptly.

7. Please remove all personal information, ensure that the data shared are in accordance with participant consent, and re-upload a fully anonymized data set.

Dear Editor,

Thank you for your kind reminder. We have thoroughly reviewed the original dataset and removed all fields that could potentially identify individual participants, ensuring that the data is fully anonymized and contains no personal information. The revised dataset has been re-uploaded in accordance with PLOS ONE's data sharing policy. Please let us know if any further adjustments are needed, and we will be happy to comply.

Dear Editor,

Thank you for your kind reminder. We have carefully reviewed our reference list and confirm that all cited works are valid and do not include any retracted articles. Therefore, no modifications to the reference list are necessary at this time. Should any issues arise in the future, we will promptly address them in accordance with the journal’s guidelines.

Response to Reviewer #1’s Comments

1. The authors mention that they have considered both positive and negative emotional dimensions, and that different types of positive and negative situations can lead to significantly different behavioral outcomes. So, how were the dimensions of positive and negative situations divided, and what specific criteria were used?

Dear Reviewer,

Thank you for your recognition of our work and your valuable suggestions. In response to your question regarding the classification criteria of positive and negative emotional dimensions, we would like to provide the following explanation:

The categorization of emotional dimensions in this study is based on the widely used bidimensional model in emotion psychology, particularly the “valence” dimension. This approach divides emotions into two independent dimensions: positive emotions (e.g., joy, confidence, satisfaction), which reflect favorable affective states, and negative emotions (e.g., anxiety, anger, frustration), which represent unfavorable emotional experiences.

For measurement, we employed the well-established Positive and Negative Affect Schedule (PANAS), which has demonstrated strong reliability and validity in emotion research. This instrument allows for the effective assessment of an individual’s positive and negative emotional levels over a specific period. We calculated separate scores for positive and negative emotions, which were then treated as two independent variables in our subsequent analyses.

2. How was the relationship between the three (emotions, unsafe psychological states, and unsafe behaviors) established, and how scientific is it?

Dear Reviewer,

Thank you for your attention to the logical structure of this study. In response to your question regarding the establishment and scientific validity of the relationships among emotion, unsafe psychological states, and unsafe behavior, we offer the following clarification:

The proposed pathway model in our study is grounded in both the theoretical framework and empirical findings of existing safety behavior research. In high-risk working environments, fluctuations in workers’ emotional states have been shown to influence their risk perception, sense of safety, and behavioral tendencies. Based on this, we hypothesized that emotions affect unsafe behavior indirectly by shaping individuals’ unsafe psychological states (e.g., anxiety, resistance, detachment).

This theoretical logic aligns with multiple prior studies and has received empirical support in our analysis. Specifically, both structural equation modeling and Bootstrap testing confirmed that unsafe psychological states serve as a significant mediator between emotional states and unsafe behavior. Therefore, we believe the proposed relationships are theoretically sound and empirically supported.

It is also worth noting that the systematic validation of the emotion–psychological state–behavior pathway in a high-risk setting such as the coal mining industry represents one of the theoretical innovations of this study.

3. Emotional fluctuations can increase miners' emotional volatility, further reducing their attention and concentration, thereby increasing the risk of operational errors and accidents. There are many factors that can influence emotions, some of which may be negative while others may be positive. But how should these emotions be understood?

Dear Reviewer,

Thank you for your recognition of our work and your valuable suggestions.

increasing the risk of operational errors and accidents. However, emotion is a complex and multidimensional psychological construct influenced by a variety of factors, including life stress, work intensity, interpersonal relationships, and even individual personality traits. Emotional states may manifest as negative experiences (e.g., anxiety, tension, anger) or as positive ones (e.g., satisfaction, pride, joy).

In this study, we adopted the two-dimensional structure of the PANAS (Positive and Negative Affect Schedule), which categorizes emotions into positive and negative dimensions. While this classification does not capture the full complexity of emotional responses (e.g., certain positive emotions like “excitement” may have negative consequences in specific situations), it provides a practical and theoretically grounded framework for understanding the general trends of emotional influence on psychological and behavioral outcomes.

Thus, we modeled and measured miners’ emotional states based on this dichotomous structure. This approach contributes to the reliability of the measurement and helps clearly delineate the pathways from emotion to psychological states and behaviors. We acknowledge that future studies could refine this framework by distinguishing specific emotional types under varying work conditions. Nevertheless, the positive/negative classification used in the current study represents a reasonable methodological and theoretical compromise at this stage of research.

4. Did the authors distinguish between emotions based on coal mine safety management and those based on other factors (including family, society, etc.) in the questionnaire? Although different emotions can have certain relationships with emotions, unsafe psychological states, and unsafe behaviors, emotions related to coal mine safety are only a part of the miners' overall emotions.

Dear Reviewer,

Thank you for your thoughtful comments and for your attention to the details of the questionnaire design. Regarding your question on whether the study distinguishes between emotions related to coal mine safety management and those from other sources, we would like to clarify as follows:

In this study, we used the revised version of the Positive and Negative Affect Schedule (PANAS) to assess participants’ subjective emotional experiences over the past week. This instrument does not explicitly differentiate the source of emotions. Therefore, the emotional data collected reflect miners’ overall affective states, rather than being limited to emotions specifically induced by coal mine safety management contexts.

We acknowledge and agree that this approach may introduce some contextual ambiguity, as it includes emotions influenced by non-work-related factors. However, given the high-risk and high-demand nature of the coal mining work environment, it is reasonable to assume that employees’ emotional states are substantially shaped by work-related conditions, such as task pressure, managerial oversight, and safety culture. Previous studies have also demonstrated that workplace emotions significantly predict safety-related behaviors. Thus, the use of PANAS in this context remains methodologically appropriate and practically meaningful.

Additionally, we have clearly acknowledged this limitation in the discussion section and suggested that future studies consider using more targeted instruments, such as work-specific emotion scales or context-sensitive assessment tools, to better capture the emotional influences specific to occupational settings and enhance the contextual sensitivity of the model.

5. The paper mentions that a cluster random sampling method was used to select 250 male workers from coal mining companies in Shandong, Henan, Shanxi, and Inner Mongolia as the subjects of the study. However, the conclusion states that the data came from a single coal mining company, which is confusing. I speculate that this might refer to a single coal mining company that includes subordinate coal mines distributed across different provinces.

Dear Reviewer,

Thank you for pointing out this potential source of confusion. The “multiple coal mining enterprises” mentioned in the manuscript actually refer to different mining sites located in various regions, all of which operate under the same corporate group and share a common legal entity. Therefore, the reference to a “single enterprise” in the conclusion indicates that all participants belonged to the same organizational system, rather than multiple independent companies.

We have clarified this point in the discussion section of the revised manuscript.

6. Section 1.2, the first letter of the title is not capitalized.

Dear Reviewer,

Thank you for your careful review. In response to your comment, we have revised the manuscript title to ensure that the initial letters are properly capitalized. Please kindly let us know if there are any other formatting issues that require further attention.

Response to Reviewer #2’s Comments

1. the sample representation and method limitations, the sample size of the data is a bit small, the universality of the conclusion is questionable, and the data presentation and analysis results need to be further modified.

Dear Reviewer,

Thank you for your thoughtful comments regarding the sample design and methodological aspects of our study. We respectfully provide the following clarifications:

Our sample consisted of 250 participants, which falls within the medium range for empirical studies in the fields of psychology and safety behavior. To enhance regional representation and typicality, we employed a cluster random sampling method that included frontline workers from coal mining regions in eastern, central, and western China (e.g., Shandong, Henan, Inner Mongolia). To minimize potential confounding effects related to differences in organizational culture and management practices, all participants were recruited from different mining sites under the same corporate group. This approach allowed us to maintain consistency in organizational context while ensuring regional heterogeneity to strengthen the robustness of our conclusions.

We also acknowledge that the homogeneity of organizational affiliation, while beneficial for controlling structural variables, may limit the external generalizability of our findings. We have explicitly discussed this limitation in Section 5.4 “Limitations and Future Directions,” and suggested that future research could expand the sampling scope to include participants from multiple companies or industries to examine the broader applicability of the proposed model.

Regarding data presentation and analysis, we have further refined the Results section to improve clarity and coherence, ensuring the accuracy of numerical reporting and logical consistency. If there are specific areas that remain unclear or problematic, we would be grateful for your guidance and will revise them carefully.

Thank you again for your valuable feedback.

2.The direction of effect of health status on unsafe behaviors was inconsistent in table 3(M1: β = 0.166; M4: β = -0.011) , please explain further.

Dear Reviewer,

Thank you for your attention to the details of our regression analysis. Regarding the concern about the inconsistent direction of the relationship between health status and unsafe behavior in Table 3, we offer the following clarification:

In Model M1, health status showed a significant positive effect on unsafe behavior (β = 0.166, p < 0.01), indicating a predictive relationship at the level of control variables. However, in Model M4, the coefficient changed direction (β = -0.011) and was no longer statistically significant (p > 0.05). This shift may be due to the inclusion of mediating variables, which likely diluted or absorbed part of the predictive path.

As health status is not a core variable in this study, and these fluctuations did not affect the main path relationships or the testing of our hypotheses, we did not elaborate on this in the manuscript. However, we are happy to revise the text and provide additional explanation should the editor deem it necessary.

3.The text mentions Fig. 2, but the manuscript does not provide an illustration, which affects the interpretation of the results.

Dear Reviewer,

Thank you for pointing out the omission of Figure 2. We have now i

---

## [Editor Report · Decision Letter 1]

12 Aug 2025

The Impact of Coal Miners' Emotions on Unsafe Behavior: A Study on the Mediated Effects with a Moderating Role

PONE-D-25-18288R1

Dear Dr. Xia,

We’re pleased to inform you that your manuscript has been judged scientifically suitable for publication and will be formally accepted for publication once it meets all outstanding technical requirements.

Kind regards,

Damiano GIRARDI

Academic Editor

PLOS ONE

Additional Editor Comments (optional):

Thank you for submitting the revised version of your manuscript to PLOS ONE. I have now carefully reviewed your responses to the reviewers’ comments and the changes you have implemented in the manuscript.

I would like to express my appreciation for the thoughtful and thorough revisions you have made. The updated version addresses the concerns raised during peer review and reflects a significant improvement in both clarity and scientific presentation.

Based on my evaluation, I find the manuscript to be scientifically sound. I am therefore pleased to recommend the manuscript for acceptance and publication in PLOS ONE.
---

## [Editor Report · Acceptance letter]

PONE-D-25-18288R1

PLOS ONE

Dear Dr. Xia,

I'm pleased to inform you that your manuscript has been deemed suitable for publication in PLOS ONE. Congratulations! Your manuscript is now being handed over to our production team.

Kind regards,

on behalf of

Dr. Damiano GIRARDI

Academic Editor

PLOS ONE